# Living with relatives offsets the harm caused by pathogens in natural populations

Hanna M Bensch, Emily A O'Connor, Charlie Kinahan Cornwallis*

Department of Biology, Lund University, Lund, Sweden

**Abstract** Living with relatives can be highly beneficial, enhancing reproduction and survival. High relatedness can, however, increase susceptibility to pathogens. Here, we examine whether the benefits of living with relatives offset the harm caused by pathogens, and if this depends on whether species typically live with kin. Using comparative meta-analysis of plants, animals, and a bacterium ($n_{species}$ = 56), we show that high within-group relatedness increases mortality when pathogens are present. In contrast, mortality decreased with relatedness when pathogens were rare, particularly in species that live with kin. Furthermore, across groups variation in mortality was lower when relatedness was high, but abundances of pathogens were more variable. The effects of within-group relatedness were only evident when pathogens were experimentally manipulated, suggesting that the harm caused by pathogens is masked by the benefits of living with relatives in nature. These results highlight the importance of kin selection for understanding disease spread in natural populations.

## Introduction

High relatedness between individuals can favour the evolution of cooperative interactions that increase reproductive success and survival (*Hamilton, 1964a*; *Hamilton, 1964b*). For example, it has been repeatedly shown that individuals can pass on their genes indirectly by providing vital resources to relatives and assisting them with tasks that are difficult to do alone, such as caring for offspring (*Alexander, 1974*; *Rubenstein and Abbot, 2017*; *West et al., 2007*). However, living with relatives can also increase susceptibility to pathogens that spread more easily among genetically similar individuals, with similar immune defences (*Anderson et al., 1986*; *Baer and Schmid-Hempel, 1999*; *Hamilton, 1987*; *Liersch and Schmid-Hempel, 1998*; *Schmid-Hempel, 1998*; *Sherman et al., 1998*). This phenomenon has been referred to as the 'monoculture effect' (*Elton, 1958*) in agricultural settings after it was observed that clonal crops were highly susceptible to disease outbreaks (*Garrett and Mundt, 1999*; *Tooker and Frank, 2012*; *Wolfe, 1985*; *Zhu et al., 2000*). More recently it has also been established that such effects occur in natural populations, with higher genetic similarity between individuals increasing rates of parasitism (*Ekroth et al., 2019*). What remains unclear is whether this translates into higher rates of mortality, or whether the benefits of living with relatives are large enough to offset the costs of increased disease risk (*Hughes et al., 2002*).

Previous research into the effects of relatedness on disease spread have been conducted on an expansive range of species including bacteria, plants, and animals. These studies have revealed remarkable variation in how changes in relatedness influence parasitism and mortality. For example, in honeybees, *Apis melifera*, high relatedness among individuals increases the risk of disease and colony death (*Tarpy et al., 2013*), whereas in Pharoah ants, *Monomorium pharaonis*, high relatedness reduces the abundance of pathogens (*Schmidt et al., 2011*). Such differences between species may in part be due to how data are collected. In some studies, relatedness and pathogens have

*For correspondence:
charlie.cornwallis@biol.lu.se

Competing interests: The authors declare that no competing interests exist.

**eLife digest** Living in a group with relatives has many advantages, such as helping with child rearing and gathering food. This has led many species to evolve a range of group behaviours; for example, in honey bee populations, worker bees sacrifice themselves to save the colony from incoming enemies.

But there are also downsides to living with family. For example, bacteria, viruses and other disease-causing pathogens will find it easier to spread between relatives. This is because individuals with the same genes have similar immune defences. So, is it better to live with relatives who can help with life's struggles or live with unrelated individuals where there is a lower chance of getting sick?

To help answer this question, Bensch et al. analysed data from 75 studies which encompassed 56 different species of plants, animals, and one type of bacteria. This showed that creatures living in family groups experienced more disease and had a higher risk of death. However, if groups had a low chance of encountering pathogens, individuals living with relatives were more likely to survive. This cancels out the disadvantages family groups face when pathogens are more common.

The analysis by Bensch et al. provides new insights into how pathogens spread in species with different social systems. This information can be used to predict how diseases occur in nature which will benefit ecologists, epidemiologists, and conservation biologists.

been experimentally manipulated, whereas in other studies relatedness and the abundances of pathogens are only observed ('observational studies'). In observational studies, results can be variable and difficult to interpret because the causality behind relationships is uncertain (*Lively et al., 2014*). For instance, a negative relationship between relatedness and the abundance of pathogens can occur either because groups of relatives are less susceptible to pathogens, or because groups of relatives die from pathogens and so are rarely observed (*Ben-Ami and Heller, 2005*; *King et al., 2011*; *Teacher et al., 2009*).

Additionally, species may vary in their susceptibility to pathogens because of differences in past selection to control disease spread among individuals (*Loehle, 1995*; *Romano et al., 2020*). In species where relatives frequently interact, selection is predicted to favour the evolution of strategies that mitigate the impacts of pathogens (*Loehle, 1995*; *Romano et al., 2020*). Limiting social interaction through group-level organisation, such as task partitioning and other mechanisms of the so-called 'social immunity', can prevent disease spread among relatives (*Camargo et al., 2007*; *Cremer and Sixt, 2009*; *Liu et al., 2019*; *Ugelvig et al., 2010*; *Waddington and Hughes, 2010*). However, whether species that typically live with kin are better able to cope with pathogens when relatives interact, compared to species that live with non-kin, is unclear.

The spread of disease through populations also depends on how variable pathogen abundances are across groups. Pathogen abundances are expected to be more variable among groups of relatives because they either contain resistant or susceptible genotypes (*Boomsma and Ratnieks, 1996*; *van Baalen and Beekman, 2006*). Groups of unrelated individuals, on the other hand, will contain a mix of susceptible and resistant genotypes, leading to more predictable pathogen abundances and rates of mortality across groups. Such differences in variation across groups of related and unrelated individuals are nevertheless predicted to depend on pathogen diversity. When there are many different pathogens, groups of relatives are more likely to be susceptible to at least one pathogen, which can reduce variation in total pathogen abundance to a level that is similar to groups of unrelated individuals (*Ganz and Ebert, 2010*; *van Baalen and Beekman, 2006*). While both increases and decreases in variation in rates of parasitism and mortality have been found in specific study species (*Ganz and Ebert, 2010*; *Johnson et al., 2006*; *Seeley and Tarpy, 2007*; *Thonhauser et al., 2016*), whether variation among groups of relatives is generally higher across species remains to be tested.

Here, we use phylogenetic meta-analysis to first examine whether the benefits of living with relatives counteract the costs of increased susceptibility to pathogens. Second, we tested if the ability to detect such effects is dependent upon the experimental manipulation of pathogens and within-group relatedness. Third, we examined if species that typically live with kin have evolved mechanisms to reduce pathogen spread among relatives compared to species that typically live with non-

kin. Finally, we investigated whether variation in the abundance of pathogens and rates of mortality is higher across groups of relatives. The influence of relatedness on mortality and pathogen abundances were quantified by extracting effect sizes (Pearson's correlation coefficients *r*) from 75 published studies across 56 species (*Supplementary file 1*—Tables S1-S3). Variation in pathogen abundances and rates of mortality were measured using a standardised effect size of variance that accounts for differences in means, the coefficient of variation ratio (CVR), which was possible to estimate for 25 species (*Supplementary file 1*—Table S4).

## Results

### Relatedness and susceptibility to pathogens

Across species, within-group relatedness had highly variable effects on rates of mortality and the abundance of pathogens (*Figure 1*. Bayesian Phylogenetic Multi-level Meta-regression (BPMM): Mean effect size [posterior mode (PM)] of Zr = 0.06, credible interval [CI] = −0.12 to 0.26, pMCMC = 0.40. *Supplementary file 1*—Table S5). Such variation was ubiquitous across all taxonomic groups and was largely independent of phylogenetic history (% of variation in Zr explained by phylogeny PM (CI) = 8.20 (0.11, 31.59). *Figure 1*; *Supplementary file 1*—Table S5). Mortality was, however, consistently higher in groups of relatives in the presence of pathogens compared to when they were absent (*Figure 2*. Zr pathogens absent versus present PM (CI) = −0.29 (−0.44, −0.12), pMCMC = 0.002. *Supplementary file 1*—Table S6). Similar effects of within-group relatedness on pathogen abundances were found (Zr pathogen abundance versus mortality PM (CI) = 0.01 (−0.10, 0.19), pMCMC = 0.51. *Supplementary file 1*—Table S6), but these were much weaker (Zr pathogen abundance PM (CI) = 0.10 (–0.10, 0.33), pMCMC = 0.31. *Supplementary file 1*—Table S6).

### Experimental studies reveal contrasting effects of relatedness in the presence and absence of pathogens

There was evidence that pathogens causally increased mortality in groups of relatives (*Figure 2*; *Supplementary file 1*—Table S7). In studies where pathogens were experimentally manipulated, groups of relatives had significantly higher mortality when pathogens were present compared to when they were absent (Zr pathogens absent versus present PM (CI) = −0.40 (−0.57, −0.21), pMCMC = 0.001. *Figure 2*; *Supplementary file 1*—Table S7). The contrasting effects of relatedness in the presence and absence of pathogens meant that overall the effect of relatedness on mortality did not significantly differ from zero (Zr pathogens present PM (CI) = 0.17 (−0.09, 0.38), pMCMC = 0.15. Zr pathogens absent PM (CI) = −0.23 (−0.49, 0.03), pMCMC = 0.11. *Figure 2*; *Supplementary file 1*—Table S7). Therefore the greater susceptibility of groups of relatives to pathogens appears to be masked by kin selected benefits of living with relatives when pathogens are rare. This may also explain why in observational studies the effect of relatedness on mortality, both in the presence and absence of pathogens, was close to zero (Zr pathogens present PM (CI) = 0.06 (−0.10, 0.27), pMCMC = 0.36. Zr pathogens absent PM (CI) = 0.10 (−0.11, 0.50), pMCMC = 0.26. *Figure 2*; *Supplementary file 1*—Table S7).

Experimental manipulations of relatedness were less important for detecting the effects of relatedness on mortality than manipulations of pathogens (*Supplementary file 1*—Table S8). Studies that experimentally manipulated within-group relatedness found similar reductions in survival in groups of relatives when pathogens were present to observational studies (Experimental studies: Zr pathogens absent versus present PM (CI) = −0.19 (−0.42, −0.08), pMCMC = 0.01. Observational studies: Zr pathogens absent versus present PM (CI) = −0.30 (−0.77, −0.10), pMCMC = 0.02. *Supplementary file 1*—Table S8).

### Responses to pathogens depend on whether species live in kin groups

Next, we tested whether species that typically live with relatives have evolved mechanisms to limit the negative effects of pathogens when within-group relatedness is high. To do this, species that typically live with relatives under natural conditions (*r* > 0.25 referred to as 'kin') were compared to those that associate with unrelated individuals (*r* < 0.25 referred to as 'non-kin'. See Materials and methods for details of data used to classify species). When pathogens were present,

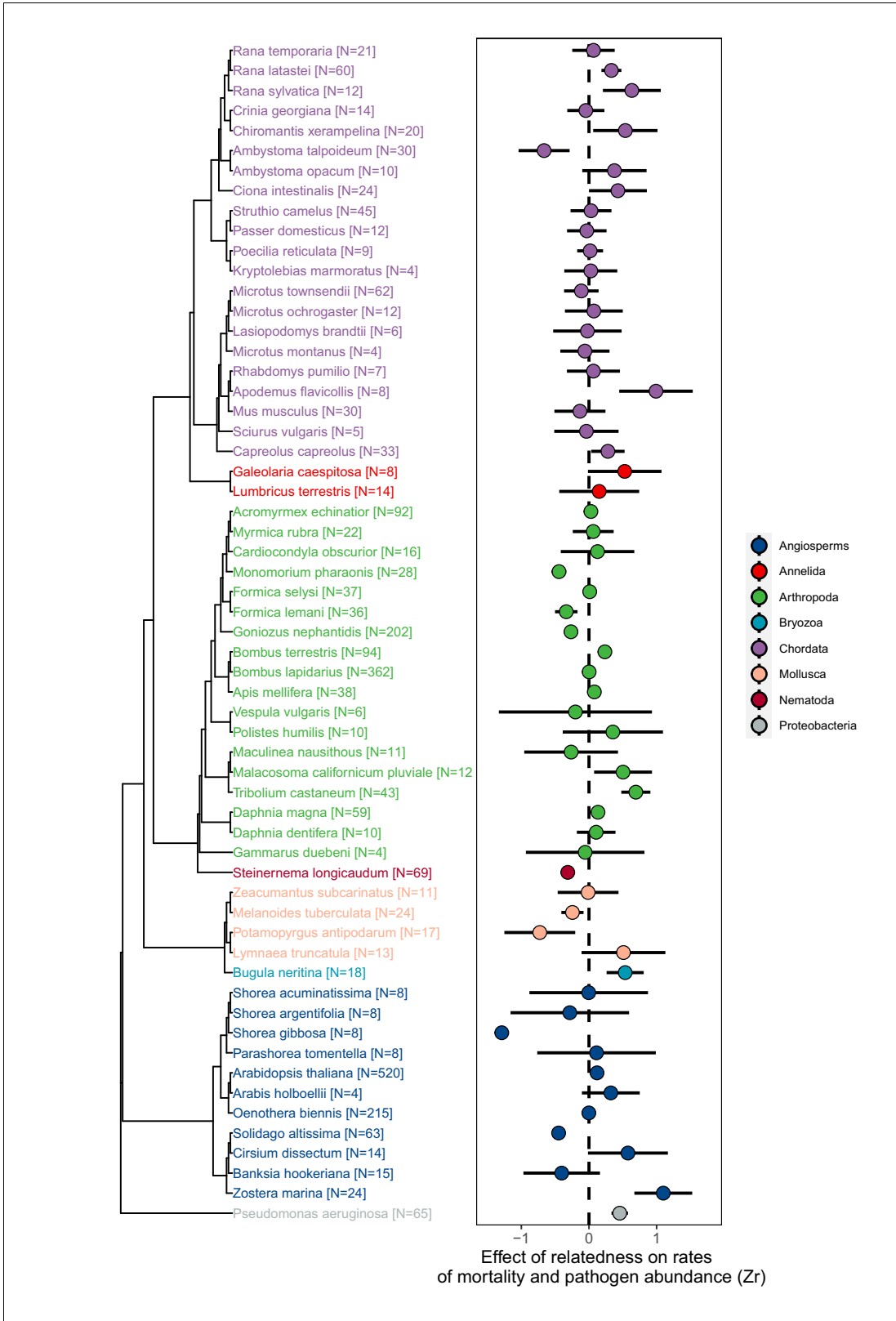

**Figure 1.** The effect of relatedness on rates of mortality and pathogen abundance across animals, plants, and bacteria. Positive effect sizes (Zr) indicate that mortality and/or pathogen abundances increase with the levels of relatedness within groups, negative values show decreases, and values of zero (dotted line) are where there was no relationship. Points represent weighted means for each species and bars are 95% confidence intervals calculated

*Figure 1 continued on next page*

*Figure 1 continued*

from the sample sizes of the number of groups studied which are given in brackets. See *Figure 1—figure supplements 1–3* for information on obtaining effect size information and testing for publication bias.

The online version of this article includes the following figure supplement(s) for figure 1:

**Figure supplement 1.** The probability that studies were of interest (A) and were included (B) in relation to the abstract relevance score (see 'Literature searches' in Materials and methods for details).

**Figure supplement 2.** Preferred reporting items for meta-analyses (PRISMA) diagram of literature search.

**Figure supplement 3.** Examining evidence of publication bias in estimates of Zr.

---

the effect of relatedness on rates of mortality did not differ between species that live with kin and non-kin (Zr kin versus non-kin pathogen present PM (CI) = 0.09 (−0.31, 0.37), pMCMC = 0.79. *Supplementary file 1*—Table S9). However, when pathogens were absent, high relatedness reduced mortality in species that live with kin, but increased mortality in species that live with non-kin (Zr kin versus non-kin pathogen absent PM (CI) = −0.57 (−1.11, 0.02), pMCMC = 0.03. *Figure 3*, *Supplementary file 1*—Table S9). For example, in the red flour beetle, *Tribolium castaneum*, and the tube worm, *Galeolaria caespitosa*, that typically interact with non-kin, mortality was two to four times higher when individuals were placed in groups of relatives compared to when individuals were unrelated (*Agashe, 2009*; *McLeod and Marshall, 2009*). These results show that species that typically associate with non-kin suffer reductions in fitness when placed in groups of relatives, but only when pathogens are rare. Conversely, species that live with kin have higher fitness in groups of relatives when pathogens are absent, but such benefits disappear when pathogens are present (Zr pathogens absent versus present PM (CI) = −0.33 (−0.53,–0.16), pMCMC = 0.001. *Supplementary file 1*—Table S9).

## Relatedness increases variance in mortality across groups, but not pathogen abundances

Variation in rates of mortality and the abundance of pathogens were influenced by relatedness in opposing ways (*Figure 4*; *Supplementary file 1*—Table S10-S13). Experimental manipulations of pathogen presence were important for detecting these effects (*Supplementary file 1*—Table S12). In observational studies, relatedness within groups had no effect on variance in mortality, either in the presence or absence of pathogens, and did not influence variance in pathogen abundances (Mortality pathogens absent: LnCVR PM (CI) = 0.44 (−0.32, 0.94), pMCMC = 0.32. Mortality pathogens present: LnCVR PM (CI) = −0.03 (−0.74, 0.65), pMCMC = 0.93. Pathogen abundance: LnCVR PM (CI) = −0.15 (−0.71, 0.50), pMCMC = 0.83. *Figure 4*, *Supplementary file 1*—Table S12). In contrast, in experimental studies mortality was more variable across groups of relatives when pathogens were present (LnCVR PM (CI) = 0.88 (0.21, 1.41), pMCMC = 0.02. *Figure 4*, *Supplementary file 1*—Table S12). The opposite pattern was true for pathogen abundances, with groups of relatives being less variable. This meant that overall, mortality was significantly more variable than the abundance of pathogens among groups of related versus unrelated individuals (LnCVR PM (CI) = 1.18 (0.56, 1.88), pMCMC = 0.001. *Figure 4*, *Supplementary file 1*—Table S12). These results suggest that pathogens spread more uniformly across groups of relatives, but effects on mortality are more variable than across groups of unrelated individuals.

## Discussion

Our analyses show that pathogens can increase rates of mortality in groups of relatives. The detrimental effects of pathogens were, however, counteracted by high relatedness reducing mortality when pathogens were rare, particularly in species that live in kin groups. Such contrasting effects of relatedness meant that experimental manipulations were crucial for detecting the costs and benefits of living with relatives when the presence of pathogens varied. Additionally, high relatedness resulted in more even abundances of pathogens across groups, but more variable rates of mortality, highlighting the importance of population genetic structure in explaining the epidemiology of diseases. We discuss these findings in relation to the environments favouring the evolution of different social systems, the mechanisms that have evolved to prevent disease spread in social groups, and the types of study system where more experimental data are required.

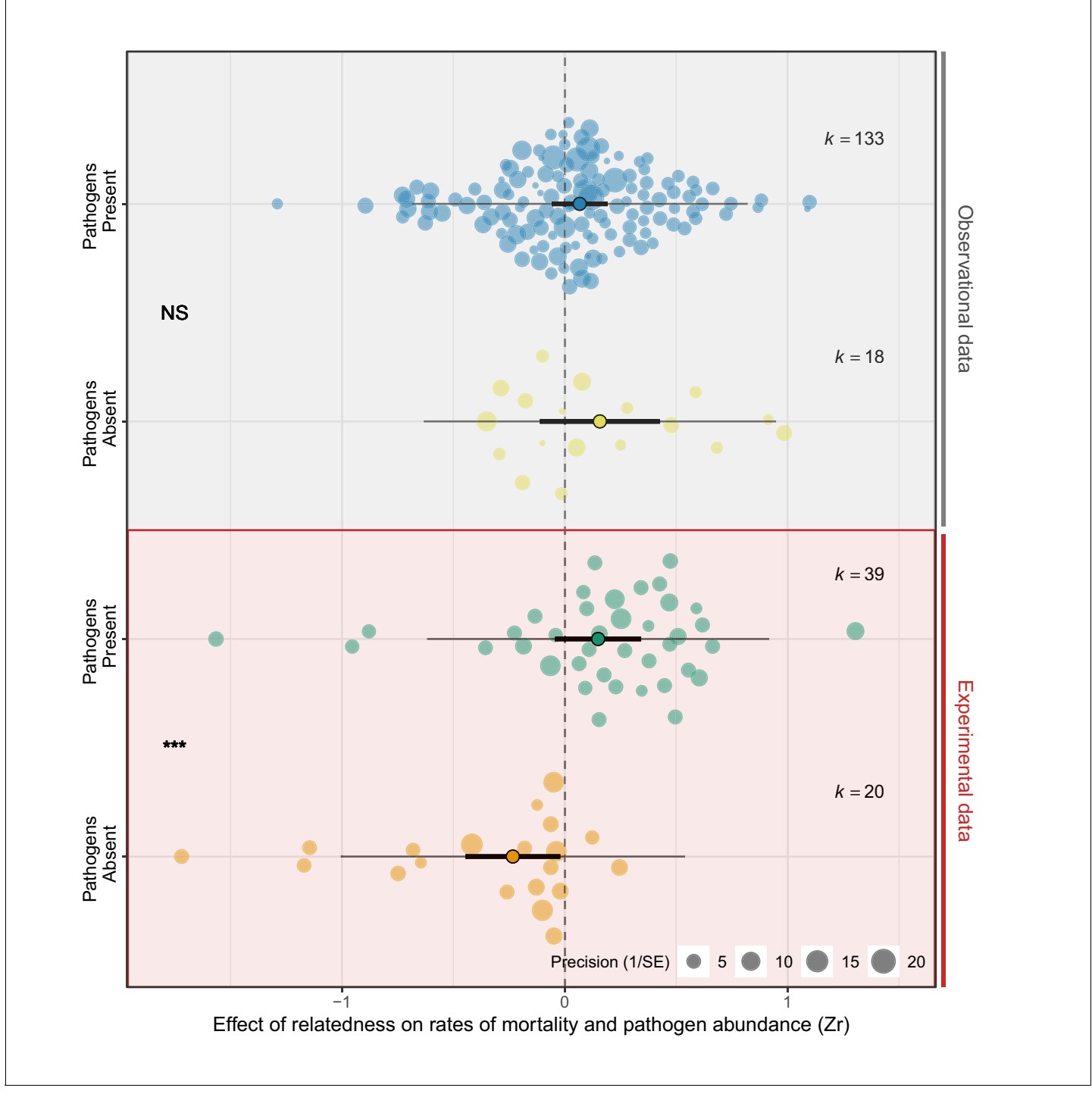

**Figure 2.** Experimental manipulations are key to detecting the effects of pathogens on groups of relatives. Positive effect sizes (Zr) indicate that mortality and/or pathogen abundances increase with the levels of relatedness within groups, negative values show decreases, and values of zero (dotted line) are where there was no relationship. Studies that experimentally manipulated pathogen presence showed that groups of relatives had higher rates of mortality when pathogens were present, but lower mortality when pathogens were absent. Points with black edges represent means, thick bars are 95% CIs, thin bars are prediction intervals, and k is the number of effect sizes. Each dot is an individual effect size and with size scaled to 1/SE (orchard plots: *Nakagawa et al., 2021*). Statistical differences are from Bayesian Phylogenetic Multi-level Meta-regressions (BPMMs) and placed mid-way between comparison groups denoted with symbols: NS = non-significant, *pMCMC < 0.05, **pMCMC < 0.01, ***pMCMC < 0.001.

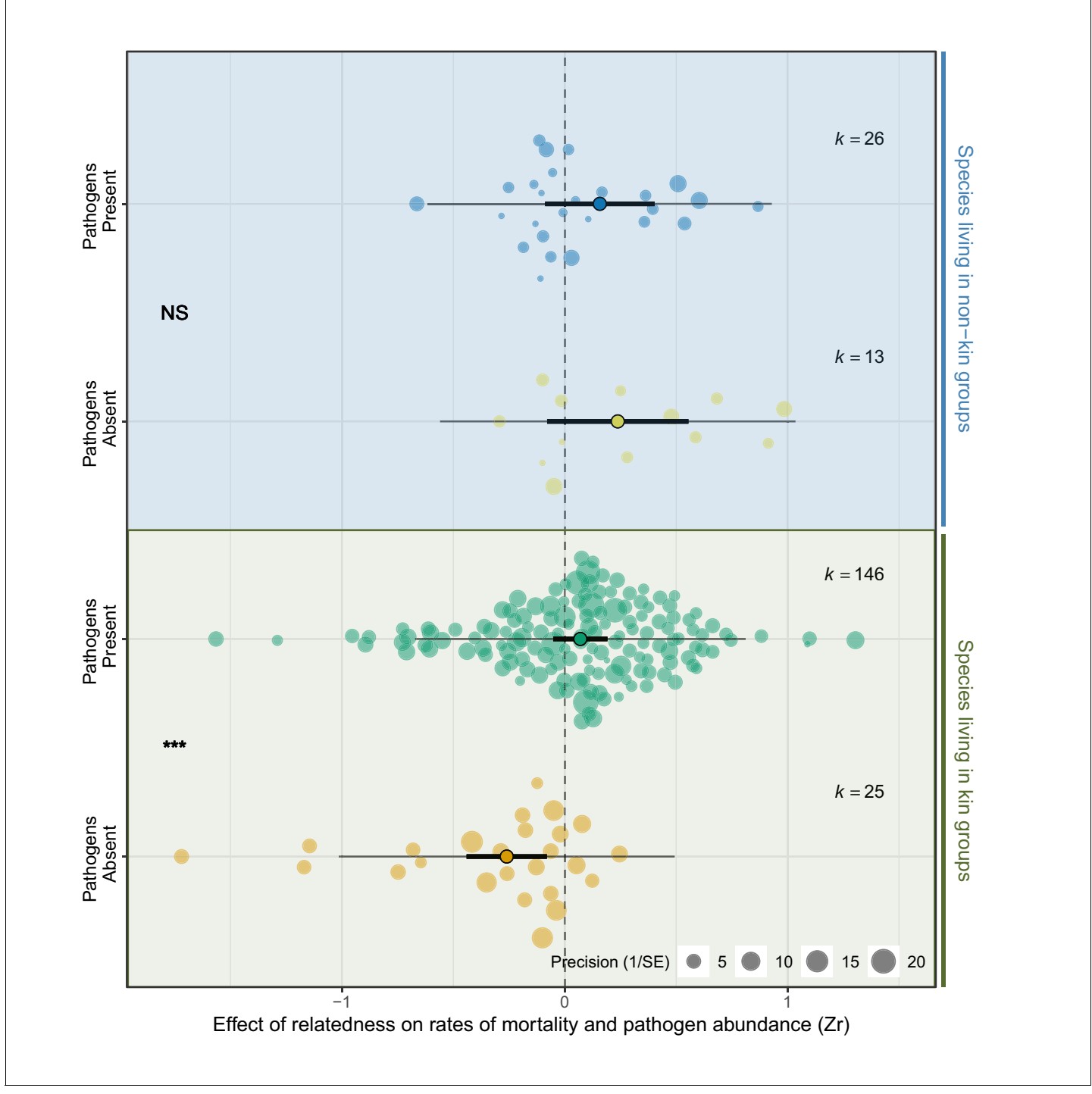

**Figure 3.** Species that live in kin groups responded differently to experimental manipulation of pathogens compared to species that live with non-kin. Positive effect sizes (Zr) indicate that mortality and/or pathogen abundances increase with the levels of relatedness within groups, negative values show decreases, and values of zero (dotted line) are where there was no relationship. When pathogens were experimentally removed species that live with kin had higher survival, which was reversed when pathogens were present. In contrast, there was no effect of relatedness on mortality when pathogens were present or absent in species that live with non-kin. The components of the orchard plots are the same as in *Figure 2*. Statistical differences are from Bayesian Phylogenetic Multi-level Meta-regressions (BPMMs) and placed mid-way between comparison groups denoted with symbols: NS = non-significant, *pMCMC < 0.05, **pMCMC < 0.01, ***pMCMC < 0.001.

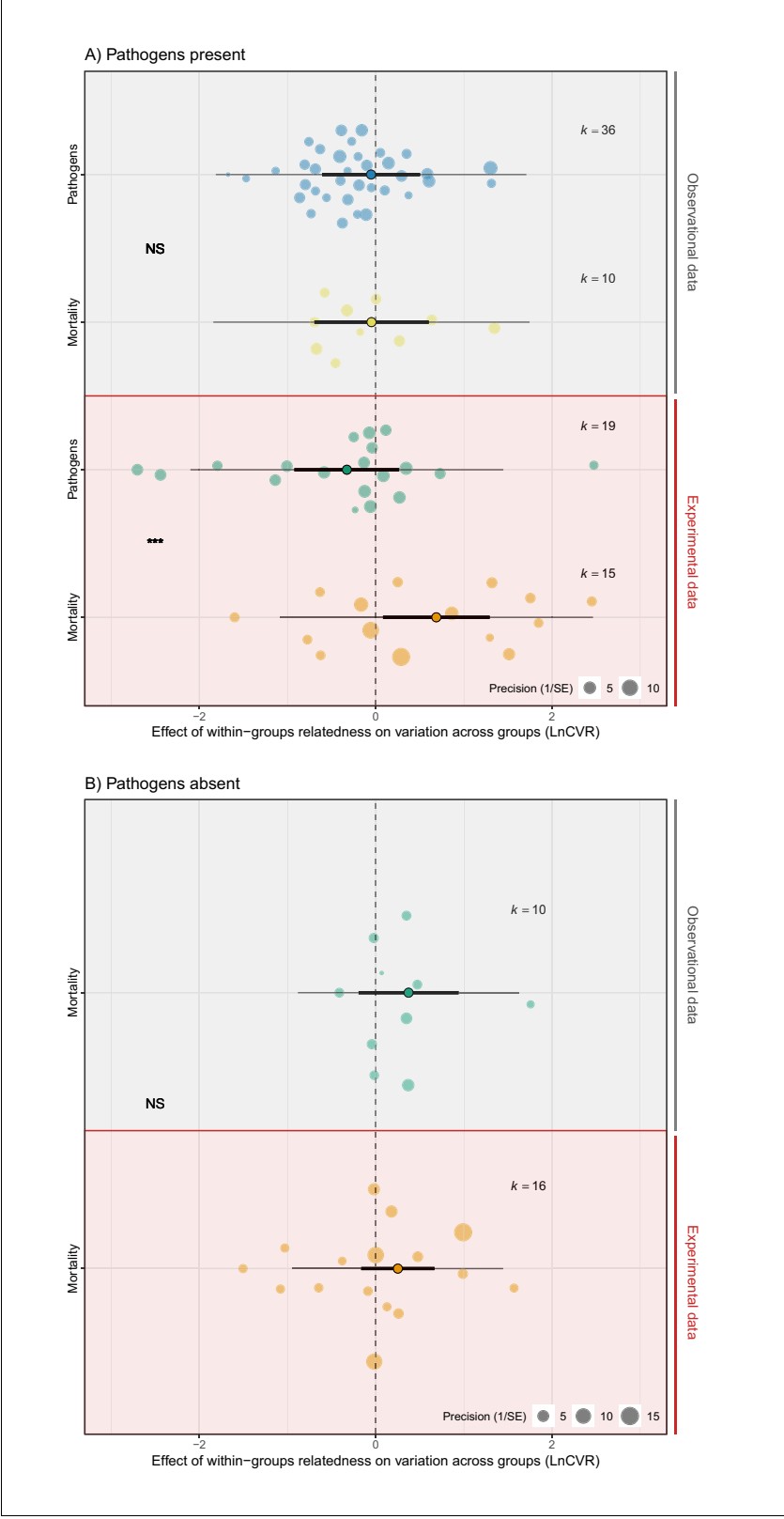

**Figure 4.** The effect of within-group relatedness on variance in mortality and pathogen abundance. Positive effect sizes (LnCVR) show that variation in rates of mortality and/or pathogen abundances across groups (accounting for mean differences – see *Figure 4—figure supplement 1* for mean variance relationship) increases with within-group relatedness, negative values show decreases in variance, and zero values show no change in variance (dotted line). (**A**) In the presence of pathogens, relatedness increased variance in mortality, but decreased variance in pathogen abundance. (**B**) When

*Figure 4 continued on next page*

*Figure 4 continued*
pathogens were absent, relatedness did not influence variance in mortality. The components of the orchard plots are the same as in *Figure 2*.
Statistical differences are from Bayesian Phylogenetic Multi-level Meta-regressions (BPMMs) and placed mid-way between comparison groups denoted
with symbols: NS = non-significant, *pMCMC < 0.05, ** pMCMC < 0.01, ***pMCMC < 0.001. *Figure 4—figure supplement 2* for examination of
publication bias.
The online version of this article includes the following figure supplement(s) for figure 4:

**Figure supplement 1.** Relationship between the mean (log) and SD (log) across studies used to estimate LnCVR.
**Figure supplement 2.** Examining evidence of publication bias in estimates of LnCVR.

The interaction between kin selected benefits and mortality caused by pathogens has important implications for our understanding of the ecological distributions of species and the evolutionary origins of different social systems. In some lineages, such as birds, cooperative species that live in families have been found to inhabit areas that are hot and dry (*Cornwallis et al., 2017*; *Jetz and Rubenstein, 2011*; *Lukas and Clutton-Brock, 2017*). This has been attributed to the benefits of cooperative offspring care being higher in environments that are challenging for independent reproduction (*Emlen, 1982*). An additional, potentially important explanation is that the costs imposed by pathogens when living with relatives may be lower in such environments (*Campbell-Lendrum et al., 2015*). Parallel arguments have been made for social insects. Species with sterile worker castes, that only evolved in groups with high levels of relatedness, are thought to have arisen in environments protected from pathogens (*Hamilton, 1987*). For example, sterile soldier castes have evolved at least six independent times in clonal groups of aphids, and the majority of these cases form galls that provide protection against pathogens (*Hamilton, 1987*; *Stern and Foster, 1996*). Escape from pathogens may therefore be a general feature governing the evolutionary origin, as well as the current ecological niches, of species living in highly related groups.

The benefits of living with relatives are predicted to generate selection for increased resistance or tolerance to disease spread (*Loehle, 1995*; *Romano et al., 2020*). Adaptations to limit pathogen transmission in kin groups have been documented in some species. For example, in leaf cutter ants, *Acromyrmex* spp., workers outside the colony, where pathogens are more prevalent, do not enter the inner colony (*Camargo et al., 2007*). Contamination of food by pathogens is also limited by workers outside the colony performing dedicated tasks, such as foraging versus waste management (*Waddington and Hughes, 2010*). Changing the organisational structure of groups or living in smaller groups can therefore increase social distancing and reduce pathogen transmission (*Loehle, 1995*; *Romano et al., 2020*; *Liu et al., 2019*).

While examples of social immunity exist, there was little evidence that species that live with kin have generally evolved mechanisms to limit the harm caused by pathogens. Species that live in kin groups suffered similar reductions in survival from pathogens to species that live with non-kin (*Figure 3*). One explanation is that individuals respond to greater pathogen pressure by forming more genetically diverse groups (*Schmid-Hempel, 1998*; *Sherman and Morton, 1988*). For example, increases in mating promiscuity under higher disease risk can lower relatedness among offspring recruited to groups (*Busch et al., 2004*; *Singh et al., 2015*; *Soper et al., 2014*). Such responses can reduce disease spread, but also weakens selection for adaptations that limit pathogen spread among related individuals. The relative costs of decreasing the effects of pathogens by reducing relatedness versus other mechanisms remains unclear, but may provide insight into why different responses to pathogens have evolved across species.

High relatedness was associated with higher and more variable rates of mortality in the presence of pathogens, but had little effect on variation in pathogen load. Such differences may arise because pathogen abundances are often weakly related to the virulence of pathogens (*Leggett et al., 2012*). Genotypes can also be equally susceptible to pathogens, but vary in their ability to clear infections, which may explain why within-group relatedness influenced mortality rates without strongly affecting variation in pathogen abundances (*Best et al., 2008*; *Howick and Lazzaro, 2014*; *Koskela et al., 2002*).

The effects of relatedness on mortality rates were only evident in experiments. There are a number of possible, non-mutually exclusive, explanations for this. It is possible that observational studies fail to capture the true effect of pathogens because of sampling biases: groups of relatives infected

with pathogens can quickly die resulting in their effects being underestimated (*Ben-Ami and Heller, 2005*; *King et al., 2011*; *Teacher et al., 2009*). The diversity and abundance of pathogens may also differ between experimental and observational studies. Although experiments often reported that pathogens were manipulated in biologically realistic ways, it is possible that pathogen abundances are generally higher in experiments leading to larger effect sizes. Additionally, experiments generally only manipulated single pathogens whereas observational studies on natural populations often involve communities of pathogens. Low pathogen diversity is predicted to increase variation across groups of relatives (*Boomsma and Ratnieks, 1996*; *van Baalen and Beekman, 2006*). The lack of an effect of relatedness on variance in mortality in observational studies may therefore be due to the diversity of pathogens being higher. In our dataset, there was only one experimental study that manipulated multiple pathogens. In *Daphnia magna* it was found that variance in parasitism was higher in groups of relatives ('clonal' versus 'polyclonal' populations), but this diminished as the number of pathogens increased (*Ganz and Ebert, 2010*). This suggests that where pathogen diversity is high, groups of relatives become increasingly susceptible to pathogens, reducing variance across groups (*Boomsma and Ratnieks, 1996*; *Parsche and Lattorff, 2018*; *van Baalen and Beekman, 2006*).

How relatedness among individuals influences pathogen spread has been investigated in a diverse range of species making our analyses possible. Nevertheless, experiments manipulating pathogen presence, abundance, and diversity across species with different ecological niches and social systems, especially those that typically associate with non-kin, remain limited. In-depth analyses comparing species with ancestrally versus derived levels of high and low relatedness will also help shed light on the importance of current versus past evolutionary responses to pathogens. We hope that our results stimulate further research in these areas which appears crucial to understanding the impact of pathogens on natural populations.

## Materials and methods

### Literature searches

A systematic literature review was performed to identify studies that have examined the relationship between within-group relatedness and rates of mortality or the abundance of pathogens. One challenge with locating relevant literature was that some studies use the term relatedness while others use the term genetic diversity. Genetic diversity encompasses studies that have examined within-individual genetic diversity (e.g. heterozygosity), as well as genetic diversity of groups. The aims of our study only relate to variation in genetic diversity of groups (relatedness). All studies where estimates of within-group genetic diversity were potentially influenced by within-individual genetic diversity were excluded (see below).

The literature search was performed using the Web of Science (WoS) including articles published up to the 27 July 2020. Searches were restricted to articles in English and the WoS categories were restricted to Behavorial Sciences, Ecology, Biology, Evolutionary Biology, Ecology, Multidisciplinary Sciences, Genetics & Heredity, Biodiversity & Conservation, Entomology, Zoology as a preliminary study (Bensch MSc thesis) showed these categories to be the ones of interest. WoS searches included the following combinations of terms in the topic field: ((('genetic diversity' OR 'genetic variability' OR 'genetic diversities') AND parasite*) OR (('genetic diversity' OR 'genetic variability' OR 'genetic diversities') AND disease*) OR (('genetic diversity' OR 'genetic variability' OR 'genetic diversities') AND pathogen*) OR (('genetic diversity' OR 'genetic variability' OR 'genetic diversities') AND survival) OR (('genetic diversity' OR 'genetic variability' OR 'genetic diversities') AND mortality) OR (relatedness AND pathogen*) OR (relatedness AND disease*) OR (relatedness AND parasite*) OR (relatedness AND mortality) OR (relatedness AND survival*) OR 'monoculture effect' OR 'Monoculture effect') AND (population* OR group* OR colony). Initial exploration of search terms included other words ('clone', 'clonal', 'social'). However, these terms inflated the number of search hits and papers with relevant data were retrieved using other terms included in our search criteria ('group', 'colony', or 'relatedness'). The search yielded a total of 4616 returns, 4615 after removing a duplicate.

To aid finding relevant papers, abstracts were downloaded and imported into R for text analysis using the quanteda package (*Benoit et al., 2018*). The frequency of words in each abstract was

calculated and used to create a relevance score according to the number of words with positive and negative interest for this study. The following words had positive associations (listed in order of priority): 'genetic', 'diversity', 'diversities', 'variation', 'relatedness', 'related', 'unrelatedness', 'unrelated', 'diverse', 'parasite', 'ectoparasite', 'ectoparasites', 'parasites', 'pathogen', 'pathogenic', 'pathogens', 'disease', 'diseases', 'diseased', 'mortality', 'survival', 'resistance', 'infection', 'infections', 'prevalence', 'tolerance', 'transmission', 'population', 'group', 'colony', 'groups', 'colonies', 'populations'. The following words had negative associations: 'human', 'humans', 'hospital', 'cancer', 'hiv', 'patients'. Papers were sorted according to their relevance scores and then manually screened to examine whether they contained data that could be used to calculate an effect size of relatedness and mortality and/or pathogen abundance. We did not include studies examining the relationship between within-group relatedness and other fitness-related measures, such as fecundity or competitive ability, because such measures are influenced by many factors other than pathogens.

We stopped screening after 2102 papers as number of new papers selected for in-depth screening decreased to less than 1% per 100 references (*Figure 1—figure supplement 1*). In addition to WoS searches, reference lists of key studies and the papers from which we extracted effect sizes were screened for relevant primary literature. PDF files of articles selected based on abstract screening were downloaded for in-depth examination of full texts. A preferred reporting items for meta-analyses diagram (*Moher et al., 2009*) of the literature screening process is shown in *Figure 1—figure supplement 2*. In total our dataset consisted of 210 effect sizes from 75 studies and 56 species (*Abdi et al., 2020*; *Agashe, 2009*; *Aguirre and Marshall, 2012a*; *Aguirre and Marshall, 2012b*; *Altermatt and Ebert, 2008*; *Anton et al., 2007*; *Baer and Schmid-Hempel, 2001*; *Baer and Schmid-Hempel, 1999*; *Ben-Ami and Heller, 2005*; *Bensch and Cornwallis, 2017*; *Bichet et al., 2015*; *Byrne and Robert, 2000*; *Byrne and Whiting, 2011*; *Cook-Patton et al., 2017*; *Cook-Patton et al., 2011*; *Crutsinger et al., 2006*; *Crutsinger et al., 2008*; *Dagan et al., 2017*; *Dagan et al., 2013*; *de Morais, 2020*; *Desai and Currie, 2015*; *de Vere et al., 2009*; *Dobelmann et al., 2017*; *Ellison et al., 2011*; *Field et al., 2007*; *Franklin et al., 2012*; *Fraser et al., 2010*; *Gamfeldt and Källström, 2007*; *Ganz and Ebert, 2010*; *Gardner et al., 2007*; *He and Lamont, 2010*; *Hoggard et al., 2013*; *Hughes and Stachowicz, 2004*; *Hughes and Boomsma, 2006*; *Hughes and Boomsma, 2004*; *Johansson et al., 2007*; *Johnson et al., 2006*; *Kapranas et al., 2016*; *Keeney et al., 2009*; *King et al., 2011*; *Kotowska et al., 2010*; *Lambin and Krebs, 1993*; *Liersch and Schmid-Hempel, 1998*; *Mattila et al., 2012*; *McLeod and Marshall, 2009*; *Mott et al., 2019*; *Neumann and Moritz, 2000*; *Page et al., 1995*; *Parker et al., 2010*; *Parsche and Lattorff, 2018*; *Pearman and Garner, 2005*; *Reber et al., 2008*; *Robinson et al., 2013*; *Schmidt et al., 2011*; *Seeley and Tarpy, 2007*; *Sera and Gaines, 1994*; *Shykoff and Schmid-Hempel, 1991*; *Siemens and Roy, 2005*; *Solazzo et al., 2014*; *Strauss et al., 2017*; *Tarpy, 2003*; *Tarpy and Seeley, 2006*; *Tarpy et al., 2013*; *Teacher et al., 2009*; *Thonhauser et al., 2016*; *Trouvae et al., 2003*; *Ugelvig et al., 2010*; *van Houte et al., 2016*; *Vanpé et al., 2009*; *Walls and Blaustein, 1994*; *Wauters et al., 1994b*; *Weyrauch and Grubb, 2006*; *Winternitz et al., 2014*; *Woyciechowski and Król, 2001*).

## Overview of study design and inclusion criteria

Studies were included if they presented data on the abundance/presence of pathogens and relatedness for four or more groups. Relatedness was estimated from breeding designs, pedigrees, and using molecular markers. A group was defined as three or more individuals as it has been shown to be sufficient for group-level defences (*Hughes and Stachowicz, 2004*). That said, only three studies used groups with three individuals (4%) with over 93% of studies using groups with five or more individuals. Some studies manipulated levels of relatedness by experimentally creating groups (referred to as 'experimental relatedness'), whereas other studies measured relatedness on already established groups (referred to as 'observational relatedness'). The presence and abundance of pathogens was also experimentally manipulated in some studies (referred to as 'experimental pathogens') whereas in others pathogens were measured without any manipulations (referred to as 'observational pathogens').

Studies on plants were included that examined the effect of pathogens and herbivores, as it has previously been argued that herbivory is equivalent to parasitism (see *Price, 1980*; *Siemens and Roy, 2005* for discussion of herbivores as pathogens). One study was included from unpublished data collected by the authors on ostriches, *Struthio camelus* (*Supplementary file 1—*Tables S19).

Studies were excluded if they were on domestic species or where there was the potential for within-individual genetic diversity, including inbreeding, to influence estimates of within-group relatedness. In some studies, inbreeding was not explicit but potentially possible (*Supplementary file 1*—Table S1). We tested the sensitivity of our results to any potential inbreeding effects by removing these effect sizes and repeating our analyses (see verification analyses; *Supplementary file 1*—Tables S14 and S15). If data of interest were missing in the text or figures, authors were contacted for supplementary data or clarification. If authors did not respond within 3 months, the effect sizes were excluded. If studies provided multiple measures of pathogen load and/or mortality, separate effect sizes were extracted. Where studies presented abundances of specific pathogens as well as total abundance of pathogens, the total was used.

## Calculating the effect size of the relationship between relatedness and rates of mortality and pathogen abundances

The relationship between within-group relatedness and mortality and/or pathogen abundance was analysed by comparing groups with high and low relatedness (relatedness as a categorical variable), or by analysing variation in average within-group relatedness as a continuous variable. Information from both types of study was used to calculate a standardised effect size of the correlation between within-group relatedness and mortality/pathogen abundance: Pearson's correlation coefficient, *r*. The statistical tests presented in studies were converted to *r* using the online meta-analysis calculator (*Morris, 2019*) and the R package 'esc' (*Lüdecke, 2019*). Measures of *r* were transformed to Zr using 'escalc' function in the R package metafor (*Viechtbauer, 2010*).

In some studies, it was not possible to obtain effect sizes directly from the statistics reported in studies, but *r* could be calculated from data presented in the text and/or figures in two ways. First, in studies where groups with high and low relatedness were compared, means ± SD of mortality or pathogen abundances were used to calculate *r*. Second, in studies where descriptive statistics (e.g. means ± SD) were reported for multiple groups that varied in relatedness, we conducted our own Pearson's correlations in R (see R script 'EffectSizeCalculations' and *Supplementary file 1*—Table S2 column 'Effect size Rscript reference'). In such cases, variation in measures of relatedness, mortality, and pathogen abundances were included by creating distributions from descriptive statistics that were sampled to create 1000 datasets. For each of these 1000 datasets, *r* was calculated and an average taken across the 1000 datasets.

## Calculating the effect size of variance in mortality and pathogen abundances across groups of related and unrelated individuals

The effect of relatedness on variance in mortality and pathogen abundances was calculated using the natural logarithm of the ratio between the coefficient of variation from groups with high and low relatedness (LnCVR: *Nakagawa et al., 2015*). LnCVR provides a standardised measure of differences in the variability of two groups accounting for differences in the means between groups. LnCVR was used because estimates of variation increased with the mean (*Figure 4—figure supplement 1*). LnCVR was calculated from studies that presented means and SDs (converted to SD if studies presented SEs or CIs) across groups when relatedness was low and high. This provides a standardised measure of the effect of relatedness on variability across groups, not within groups (SDs were from across groups, not individuals).

## Data on study characteristics

For each effect size extracted, we collected information on: (1) whether pathogens were present or absent; (2) whether pathogens were experimentally manipulated; (3) whether relatedness was experimentally manipulated; (4) the method used for measuring relatedness (pedigree or molecular markers); and (5) whether pathogen abundance or mortality were measured (where survival estimates were presented, the sign of the effect size was reversed). If there was no mention of pathogens in the paper, then pathogens were assumed to be present when studies were conducted in the field and absent if conducted in the laboratory.

## Data on species characteristics

For all species in our dataset we searched for whether they typically associate with kin ('kin') or not ('non-kin') during the life stage that effect sizes were measured. Species were categorised as kin if they lived in groups where $r$ was estimated to be equivalent to 0.25 or higher and 'non-kin' if they live in groups where relatedness was estimated to be lower than 0.25 (*Supplementary file 1*—Table S4). Three sources of information were used to estimate levels of relatedness among individuals: (1) estimates of relatedness acquired either directly from molecular genetic analyses or records of groups of individuals with known relatedness; (2) information on the mating system; and (3) typical dispersal patterns, as low dispersal from groups increases relatedness. The relevant information was collected using Google Scholar including each species name combined with 'genetic diversity', 'relatedness', and 'group' as search terms to collect measures of within-group relatedness; 'mating system' and 'paternity' for information on mating system; and 'dispersal' and 'philopatry' for information on dispersal. The categorisation of each species as kin or non-kin along with evidence and the list of literature to support these classifications can be found in *Supplementary file 1*—Table S4 (*Abdi et al., 2020*; *Aguirre and Marshall, 2012a*; *Aguirre and Marshall, 2012b*; *Amiri et al., 2017*; *Anton et al., 2007*; *Arnaud, 1999*; *Avise and Tatarenkov, 2015*; *Barrett et al., 2005*; *Bee, 2007*; *Beermann et al., 2015*; *Ben-Ami and Heller, 2005*; *Bryja et al., 2008*; *Byrne and Robert, 2000*; *Byrne and Whiting, 2011*; *Chapuisat et al., 2004*; *Croshaw et al., 2009*; *M. Crutsinger et al., 2008*; *Dagan et al., 2013*; *Dean et al., 2006*; *de Morais, 2020*; *de Vere, 2007*; *de Vere et al., 2009*; *Dobelmann et al., 2017*; *Edenbrow and Croft, 2012*; *Farentinos, 1972*; *Ficetola et al., 2010*; *Field et al., 2007*; *Franklin et al., 2012*; *Fredensborg et al., 2005*; *Gamfeldt and Källström, 2007*; *Gardner et al., 2007*; *Getz et al., 1993*; *Goldberg et al., 2013*; *Goulson et al., 2002*; *Goymann, 2009*; *Graham, 1941*; *Griffin, 2012*; *Haag et al., 2002*; *He et al., 2004*; *Head and Yu, 2004*; *Heppleston, 1972*; *Heske and Ostfeld, 1990*; *Hoffmann et al., 2003*; *Hoggard et al., 2009*; *Hughes and Stachowicz, 2004*; *Johnson, 2007*; *Johnson et al., 2006*; *Kapranas et al., 2016*; *Kawamura et al., 1991*; *Keeney et al., 2009*; *Kelly et al., 1999*; *Keough, 1989*; *Keough and Chernoff, 1987*; *Kimwele and Graves, 2003*; *King et al., 2011*; *Kozakiewicz et al., 2009*; *König, 1993*; *Lambin and Krebs, 1991*; *Laurila and Seppa, 1998*; *Lepais et al., 2010*; *Liker et al., 2009*; *Liu et al., 2013*; *Mackiewicz et al., 2006*; *McLeod and Marshall, 2009*; *Meling-lópez and Ibarra-Obando, 1999*; *Myers et al., 2011*; *Oettler and Schrempf, 2016*; *Osváth-Ferencz et al., 2017*; *Pai and Bernasconi, 2007*; *Pietrzak et al., 2010*; *Platt et al., 2010*; *Reusch et al., 1999*; *Rice et al., 2009*; *Rock et al., 2007*; *T. Russell et al., 2004*; *Schmid-Hempel and Crozier, 1999*; *Schmid-Hempel and Schmid-Hempel, 2000*; *Schmidt et al., 2011*; *Schmidt et al., 2016*; *Schradin et al., 2010*; *Schrempf et al., 2006*; *Seppä and Walin, 1996*; *Seppä et al., 2009*; *Shapiro and Dewsbury, 1986*; *Siemens and Roy, 2005*; *Simeonovska-Nikolova, 2007*; *Solomon et al., 2004*; *Stürup et al., 2014*; *Sutcliffe, 2010*; *Svane and Havenhand, 1993*; *Tarpy, 2003*; *Tatarenkov et al., 2007*; *Thonhauser et al., 2016*; *Trouvae et al., 2003*; *Vanpé et al., 2009*; *Verrell and Krenz, 1998*; *Walck et al., 2001*; *Waldman, 1982*; *Walls and Blaustein, 1994*; *Wauters et al., 1990*; *Wauters and Dhondt, 1992*; *Wauters et al., 1994a*; *Zenner et al., 2014*). We also collected data on whether species always lived in social groups ('obligately social') or whether species were only social during specific life stages ('periodically social'). However, it was not possible to analyse these data as experimental manipulations of pathogens, a key factor influencing the relationship between relatedness and mortality and pathogen abundances, were only performed for one periodically social species (*Rana latastei*).

## Data limitations

Our dataset highlighted that there are several key variables where data are limited and where further empirical work would be extremely useful. In particular, information on the following is currently limited: (1) species that typically live with non-kin ($r$: kin = 41, non-kin = 15. *LnCVR*: kin = 18, non-kin = 7); (2) studies that quantify the effect of relatedness on rates of mortality in the *absence* of pathogens, particularly under natural conditions. Out of 75 studies, pathogens were excluded in 16 laboratory studies and no studies tried to explicitly exclude pathogens under field conditions. For *LnCVR*, pathogens were only excluded in seven laboratory studies out of a total of 32 studies; and (3) variation across groups in rates of mortality and pathogen abundance (out of 210 mean effect sizes, variance could only be examined in 106).

## Statistical analysis

### General techniques

Data were analysed using Bayesian Phylogenetic Multi-level Meta-regressions (BPMM) with Markov chain Monte Carlo (MCMC) estimation and Gaussian error distributions in R package MCMCglmm (*Hadfield, 2010*). Data points were weighted by the inverse sampling variance associated with each of the effect size using the 'mev' term in MCMCglmm.

$$Variance \; r = 1/n - -3$$

$$Variance \; LnCVR = \begin{array}{l} \frac{s^2 L}{nL\dot{x}^2 L} + \frac{1}{2(nL-1)} - 2\rho \ln \dot{x}L, lnsL \sqrt{\frac{s^2 L}{nL\dot{x}^2 L} \frac{1}{2(nL-1)}} \\ + \frac{s^2 R}{nR\dot{x}^2 R} + \frac{1}{2(nR-1)} - 2\rho \ln \dot{x}R, lnsR \sqrt{\frac{s^2 R}{nR\dot{x}^2 R} \frac{1}{2(nR-1)}} \end{array}$$

where $n$ corresponds to the number of groups, $L$ and $H$ are groups with low and high relatedness, respectively. Unfortunately, the difference in relatedness between low and high relatedness treatments could not be included as a moderator in analyses because exact estimates of relatedness were not always given (e.g. monogamous versus polyandrous breeders) or comparable across studies (e.g. estimates of relatedness from molecular markers do not always equate to relatedness estimates from pedigrees/breeding designs).

The non-independence of data arising from multiple effect sizes per study were modelled by including study as a random effect. In one study (*Reber et al., 2008*), there were three relatedness treatment groups (low, intermediate, and high) allowing effect sizes between low and intermediate, and high and intermediate to be calculated. However, we excluded comparisons with the intermediate treatment to avoid non-independence of effect sizes within studies (*Noble et al., 2017*). The non-independence of data arising from shared ancestry were modelled by including a phylogenetic variance-covariance matrix of species relationships as a random effect. The phylogenetic variance-covariance matrix was created from hierarchical taxonomic classifications using the 'as.phylo' function in the R package 'ape' (see *Figure 1*). We also created a phylogeny using information from the open tree of life (*Rees and Cranston, 2017*) using the R package 'rotl' (*Michonneau et al., 2016*). This produced a tree that was extremely similar, but several mollusc species were missing and we therefore used the tree created from taxonomy. Branch lengths were estimated using Grafen's method (*Grafen, 1989*) implemented in the R package 'ape' (*Paradis, 2012*).

Fixed effects were considered significant when 95% credible intervals did not overlap with 0 and pMCMC were less than 0.05 (pMCMC = percentage of iterations above or below a test value correcting for the finite sample size of posterior samples). Default fixed effect priors were used (independent normal priors with zero mean and large variance [$10^{10}$]) and for random effects inverse gamma priors were used ($V = 1$, nu = 0.002). Each analysis was run for 1,100,000 iterations with a burn-in of 100,000 and a thinning level of 1000. Convergence was checked by running each model three times and examining the overlap of traces, levels of autocorrelation, and testing with Gelman and Rubin's convergence diagnostic (potential scale reduction factors <1.1).

### Specific analyses

Two sets of analyses were conducted, one on the effect of relatedness on mean rates of mortality and pathogen abundances ($Zr$) and one on variances ($LnCVR$). All models were fitted with a Gaussian error distribution, study, species, and phylogeny as random effects and each data point was weighted by the inverse sampling variance. Six analyses of mean effect sizes were conducted that had the following fixed effects (moderators): (1) intercept-only model to test whether overall relatedness increased susceptibility to pathogens and increased mortality; (2) three-level factor of whether mortality was measured in the presence of pathogens, mortality was measured in the absence of pathogens, or whether the abundance of pathogens was examined (referred to here as 'fitness measure'); (3) four-level factor of the effect of presence and absence of pathogens in experimental versus observational studies; (4) four-level factor of the effect of experimentally manipulating or observing relatedness in the presence and absence of pathogens; and (5) eight-level factor of the effect of living with kin and non-kin in the presence and absence of pathogens in experimental and

observational studies. All analyses were repeated for *LnCVR* apart from five, as variance estimates were only available for seven species that live with non-kin.

## Verification analyses

We checked the robustness of our results to potential inbreeding effects (Zr and LnCVR: *Supplementary file 1*—Tables S14 and S15), whether studies were conducted in laboratories or under natural conditions (Zr and LnCVR: *Supplementary file 1*—Tables S16 and S17), and the type of statistical tests used in studies (Zr: *Supplementary file 1*—Table S18). To check for effects of potential inbreeding, we repeated analysis 4 (see above) removing data points where there was any chance of inbreeding (see *Supplementary file 1*—Table S1 for effect size details. See *Supplementary file 1*—Tables S14-16 for re-analysis). There was a large overlap in whether studies were conducted in laboratories and whether they were observational or experimental: All studies conducted in laboratories were experimental whereas for observational studies 141 effect sizes were from field studies and 23 from laboratory studies. To check for laboratory effects, we therefore restricted data to observational studies and tested if effect sizes differed between laboratory and field studies (*Supplementary file 1*—Tables S16 and S17). To examine the influence of the type of statistical tests used in studies (number of different analysis techniques = 15), we included 'analysis technique' as a random effect in our main model (analysis 4 above: see R script 'ZrModels' M9). The main conclusions of our study remained unchanged and quantitatively similar in all verification analyses (*Supplementary file 1*—Tables S14-S18).

## Testing for publication bias

Publication bias across studies was checked using funnel plot visualisation and Egger's regression (*Egger et al., 1997*). Egger's regressions of both Zr and LnCVR were performed by including the inverse sampling variance as a covariate in our full model (analysis 4 above: see R script 'Publication-Bias'). In both analyses, the slope of the inverse sampling variance was not significantly different from zero (BPMM: inverse sampling variance on Zr CI = $-0.03$ to $0.01$ and LnCVR CI = $-0.04$ to $0.12$) and funnel plots of residuals were also generally symmetrical (*Figure 1—figure supplement 3*; *Figure 4—figure supplement 2*), indicating there was little evidence of publication bias.

# Acknowledgements

This research was funded by the Knut and Alice Wallenberg Foundation (Wallenberg Academy fellowship number 2018.0138) and the Swedish Research Council (grant number 2017–03880). We are very grateful to Dan Noble, Dieter Ebert, Christian Rutz, Jacobus Boomsma, and an anonymous reviewer for comments on the manuscript.

# Additional information

### Funding

| Funder | Grant reference number | Author |
| --- | --- | --- |
| Knut och Alice Wallenbergs Stiftelse | 2018.0138 | Charlie Kinahan Cornwallis |
| Vetenskapsrådet | 2017-03880 | Charlie Kinahan Cornwallis |

The funders had no role in study design, data collection and interpretation, or the decision to submit the work for publication.

### Author contributions

Hanna M Bensch, Conceptualization, Data curation, Formal analysis, Investigation, Visualization, Methodology, Writing - review and editing; Emily A O'Connor, Conceptualization, Data curation, Investigation, Methodology, Writing - review and editing; Charlie Kinahan Cornwallis, Conceptualization, Resources, Data curation, Software, Formal analysis, Supervision, Funding acquisition,

Validation, Investigation, Visualization, Methodology, Writing - original draft, Project administration, Writing - review and editing

### Author ORCIDs
Hanna M Bensch ![ORCID] https://orcid.org/0000-0002-8449-9843
Emily A O'Connor ![ORCID] https://orcid.org/0000-0001-8702-773X
Charlie Kinahan Cornwallis ![ORCID] https://orcid.org/0000-0003-1308-3995

### Decision letter and Author response
Decision letter https://doi.org/10.7554/eLife.66649.sa1
Author response https://doi.org/10.7554/eLife.66649.sa2

## Additional files

### Supplementary files
• Supplementary file 1. Supplementary Tables S1-S19 and R session information in html format. Tables S1-19 can be found in excel format at the open science framework DOI 10.17605/OSF.IO/Q3ANE. Table S1: The references screened for effect sizes of within-group relatedness on mortality and pathogen abundance and whether they were included in analyses. Table S2: Data used for analysis of the effect of within-group relatedness on mortality and pathogen abundance ($r$). Table S3: Data on whether species typically live with kin or non-kin. Table S4: Data used for analysis of the effect of within-group relatedness on variance in mortality and pathogen abundance (LnCVR). Table S5: The overall effect of within-group relatedness on mortality and pathogen abundance on Zr. Table S6: The effect of pathogen presence and whether mortality or pathogen abundance was measured on Zr. Table S7: The effect of pathogen manipulation on Zr. Table S8: The effect of relatedness manipulations and pathogen presence on Zr. Table S9: The effect of kin structure and pathogen presence measures on Zr. Table S10: The overall effect of within-group relatedness on variance (LnCVR) mortality and pathogen abundances. Table S11: The effect of pathogen manipulation on LnCVR. Table S12: The effect of pathogen manipulation and whether mortality or pathogen abundance was measured on LnCVR. Table S13: The effect of relatedness manipulations and pathogen presence on LnCVR. Table S14: The effect of kin structure and pathogen presence measures on Zr excluding potential effects of inbreeding. Table S15: Effect of parasite manipulations and fitness measures on LnCVR excluding potential effects of inbreeding. Table S16: Laboratory effects on Zr. Table S17: Laboratory effects on LnCVR. Table S18: The sensitivity of analyses on Zr to the statistical techniques used in primary studies. Table S19: Data on within-group relatedness on mortality in ostrich chicks.

• Transparent reporting form

### Data availability
All data, code and supplementary information are available at the open science framework (OSF): http://doi.org/10.17605/OSF.IO/Q3ANE.

The following dataset was generated:

| Author(s) | Year | Dataset title | Dataset URL | Database and Identifier |
|---|---|---|---|---|
| Bensch HM, O'Connor E, Cornwallis CK | 2021 | Living with relatives offsets the harm caused by pathogens in natural populations | https://osf.io/q3ane/ | Open Science Framework, 10.17605/OSF.IO/Q3ANE |

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
