## [Decision Letter]

**Acceptance summary:**

Group living may be beneficial for many reasons, but has costs in terms of increased rates of parasitism, in particular if group members are highly related. In this meta-analysis, many original studies on questions related to parasitism, relatedness and group living are brought together in one unifying framework. The authors conclude that living in groups can indeed facilitate the spread of infectious diseases, but that these costs can be outweighed by the benefits of group living.

**Decision letter after peer review:**

Thank you for submitting your article "Living with relatives offsets the harm caused by pathogens in natural populations" for consideration by *eLife*. Your article has been reviewed by two peer reviewers, one of whom is a member of our Board of Reviewing Editors, and the evaluation has been overseen by Christian Rutz as the Senior Editor. The following individual involved in the review of your submission has agreed to reveal their identity: Jacobus J Boomsma (Reviewer #2).

The reviewers have discussed their reviews with one another, and the Reviewing Editor has drafted this decision letter to help you prepare a revised submission.

This meta-analysis addresses a double-edged sword in evolutionary biology. Group living may be beneficial for many reasons, but has costs in terms of increased rates of parasitism. Furthermore, if groups are highly related, parasites that are genetically able to infect one member of the group, may be able to infect all of them, putting the entire group at risk. In the present meta-analysis, many original studies on questions related to parasitism, relatedness and group living are brought together in one unifying framework. The authors find that, indeed, group living can facilitate the spread of infectious diseases. However, they also find that the negative effects of disease can be overcompensated by the benefits of being social. The authors stress that experimental studies are necessary to disentangle these effects. The study is of a high standard and well-conducted. The take home message is clear and of general interest.

The referees have a number of useful suggestions to help you improve the manuscript. From our perspective, all of them seem easy to incorporate and there are no major issues. A detailed list is given below.

*Reviewer #1 (Recommendations for the authors):*

I was surprised to read that among the key words used at different stages of the refinement process for the literature search, term like: clone, clonal, social,.… did not show up.

A focus of the study is on effects on mortality. However, many pathogens are hardly lethal, but reduce other fitness relevant traits strongly, such as competitive ability, work performance, sexual attractiveness and fecundity.

Hand-picked examples of individual studies read nice and are entertaining, but they do not support the overall conclusion, but rather bias the observer. I suggest to leave them out, unless of specific relevance. E.g. the example of the frog and house mouse at the beginning of the result section, is not needed. Likewise the Tribolium and the worm examples (line 149).

*Reviewer #2 (Recommendations for the authors):*

– 38. I have not checked but remember that Baer and Schmid-Hempel (you cite these papers) had a more direct experimental manipulation approach than Liersch and Schmid-Hempel because they inseminated bumblebee queens with either 1 or 4 ejaculates. Worth citing here too?

– 44. Why is extinction risk relevant? Natural selection only sees individual inclusive fitness. Rephrase to avoid ambiguity?

– 57-60. Unclear sentence – rephrase.

– 67. It was Sylvia Cremer who developed the field of 'social immunity' in the first decade of this century. I think that pioneering work should be cited here. You have one of her papers (Ugelvig et al), but possibly the review by Cremer and Sixt is more general.

– 79. Replace the second 'to' by 'as in' to make sentence clearer?

– 106. It seems rather excessive to use three decimals here.

– 191-193. As in 44, the phrasing seems to have a slight and I'm sure unintentional group-selection slant. Can some rewording repair that?

– 205-207In Acromyrmex individuals are non-totipotent caste members and they do not have a 'choice' between living with relatives of non-relatives. Further, relatedness in Acromyrmex colonies is very low for ant standards, so even though you have them in the high relatedness (>0.25) category, they would be in the low-relatedness category when you would focus on social Hymenoptera only. The Hughes papers that you cite have precise relatedness estimates I believe. Somehow the phrasing of your text does not seem to capture these peculiarities.

– The numerical order of the supplementary tables seems rather haphazard. Should they not be numbered in the order they are presented in the text?

---

## [Author Response]

Reviewer #1 (Recommendations for the authors):I was surprised to read that among the key words used at different stages of the refinement process for the literature search, term like: clone, clonal, social,.… did not show up.

We initially tried combinations of search terms to identify those that increased the number of relevant papers without inflating the number of search hits. Relevant papers that used clone and clonal also used genetic diversity and monoculture and were therefore captured by our searches, but returned many hits from agricultural studies which we wanted to exclude. The term social was too broad, dramatically increasing our search hits. Relevant papers that used social also used the terms group, colony or relatedness and so we are confident that are search criteria captured relevant papers associated with these extra key words.

We have now clarified this point in the methods, section ‘Literature searches’ which reads:

“Initial exploration of search terms included other words (‘clone’,’clonal’,’social’). However, these terms inflated the number of search hits and papers with relevant data were retrieved using other terms included in our search criteria (‘group’, ‘colony’ or ‘relatedness’)” (Lines 276-279).

A focus of the study is on effects on mortality. However, many pathogens are hardly lethal, but reduce other fitness relevant traits strongly, such as competitive ability, work performance, sexual attractiveness and fecundity.

We agree that pathogens can have varied effects on different fitness related traits beyond mortality that are likely important for individuals. However, the relationship between within-group relatedness and measures such as fecundity can vary for many reasons other than pathogens making interpretation complicated. For example, local mate competition theory predicts that investment in sexual traits varies with local relatedness among individuals. Furthermore, traits such as sexual attractiveness and competitive ability are not easily interpreted in some species. In social insects, the relationship between relatedness and average attractiveness or competitive ability of workers does not have clear meaning to the hypotheses we tested. We therefore decided to stick with measures (mortality and pathogen abundance) that clearly related to the hypotheses we tested and that were comparable across all species.

We have now clarified in the methods why we excluded studies using other proxies of fitness. It reads:

“We did not include studies examining the relationship between within-group relatedness and other fitness related measures, such as fecundity or competitive ability, because such measures are influenced by factors other than pathogens” (Lines293-295).

Hand-picked examples of individual studies read nice and are entertaining, but they do not support the overall conclusion, but rather bias the observer. I suggest to leave them out, unless of specific relevance. E.g. the example of the frog and house mouse at the beginning of the result section, is not needed. Likewise the Tribolium and the worm examples (line 149).

We are grateful for the suggestion. We did not intend to bias the observer, but to give the reader greater insight into the primary studies. Meta-analyses can help cut through the species-specific idiosyncrasies allowing more general interpretation, but one downside is that the reader is left wondering what kind of studies were included. For this reason, we believe some examples can increase interpretability. Nevertheless, we have reduced these (example at the start of the results removed) in accordance with the referee’s suggestion.

Reviewer #2 (Recommendations for the authors):– 38. I have not checked but remember that Baer and Schmid-Hempel (you cite these papers) had a more direct experimental manipulation approach than Liersch and Schmid-Hempel because they inseminated bumblebee queens with either 1 or 4 ejaculates. Worth citing here too?

We have now added this citation.

– 44. Why is extinction risk relevant? Natural selection only sees individual inclusive fitness. Rephrase to avoid ambiguity?

We have now rephrased this, which reads:

“What remains unclear, however, is whether this translates into higher rates of mortality, or whether the benefits of living with relatives are large enough to offset the costs of increased disease risk” (Lines 49-51).

– 57-60. Unclear sentence – rephrase.

We have revised this sentence, which reads:

“For instance, a negative relationship between relatedness and the abundance of pathogens can occur either because groups of relatives are less susceptible to pathogens, or conversely because groups of relatives frequently die from pathogens and so rarely observed” (Lines 62-65).

– 67. It was Sylvia Cremer who developed the field of 'social immunity' in the first decade of this century. I think that pioneering work should be cited here. You have one of her papers (Ugelvig et al), but possibly the review by Cremer and Sixt is more general.

Thank you for the suggestion. We now cite Ugelvig et al. as well as Cremer and Sixt.

– 79. Replace the second 'to' by 'as in' to make sentence clearer?

Addressed.

– 106. It seems rather excessive to use three decimals here.

Agreed! The manuscript is written in R markdown so results can be directly inserted into text to increase reproducibility, but sometimes the formatting slips through the net. We have now changed the results to 2 decimals throughout.

– 191-193. As in 44, the phrasing seems to have a slight and I'm sure unintentional group-selection slant. Can some rewording repair that?

We have now rephrased this sentence which reads:

“One explanation is that individuals respond to greater pathogen pressure by forming more genetically diverse groups” (Lines 210-211).

– 205-207: In Acromyrmex individuals are non-totipotent caste members and they do not have a 'choice' between living with relatives of non-relatives. Further, relatedness in Acromyrmex colonies is very low for ant standards, so even though you have them in the high relatedness (>0.25) category, they would be in the low-relatedness category when you would focus on social Hymenoptera only. The Hughes papers that you cite have precise relatedness estimates I believe. Somehow the phrasing of your text does not seem to capture these peculiarities.

This example was to illustrate that spatial segregation and task partitioning within groups can reduce the spread of pathogen transmission among group members. While Acromyrmex have lower (derived) relatedness than some other social insects, they still have relatively high relatedness compared to the other species in our dataset (E.g. Acromyrmex octospinosus: r = ca. 0.33, Boomsma et al. 1999. Acromyrmex echinatior r = 0.380±0.042 Sumner et al. 2004). We have tried to rephrase this section to clarify the point of this example.

It reads:

“For example, in leaf cutter ants, Acromyrmex spp, workers outside the colony, where pathogens are more prevalent, do not enter the inner colony (Camargo et al., 2007). Contamination of food by pathogens is also limited by workers outside the colony performing dedicated tasks, such as foraging versus waste management (Waddington and Hughes, 2010)” (Lines 200-204).

– The numerical order of the supplementary tables seems rather haphazard. Should they not be numbered in the order they are presented in the text?

Thank you for pointing this out. The supplementary tables are now numbered in the order they appear in the text.